# Relations between Parenting Style and Parenting Practices and Children's School Achievement

**Dimitra Tsela [1,2,*], Rosie Drosou Tsela [3,*] and Ignacio González López [1]**

[1] Department of Education, University of Córdoba, 14071 Córdoba, Spain
[2] Department of Education, University of Aristotle, 54124 Thessaloniki, Greece
[3] Departments of Education and Psychology, University of Bolton, Bolton BL3 5AB, UK
[*] Correspondence: dimitra_tsela@hotmail.gr (D.T.); tsela@acadimia.com (R.D.T.)

**Abstract:** This study examines the relationship between parenting patterns and children's school achievements in Greek society. Parenting practices and parenting style are two child-rearing dynamics which were selected to assess children's school achievements in this study. A total of 101 participants who have at least one child attaining elementary school and who reside in Greece answered an online questionnaire. In line with previous findings, the aim of this study is to examine associations between parenting and primary school students' achievements in Greece. The initial assumption was that both parenting practices and parenting style are associated with academic achievement. Thus, the primary hypotheses under examination in this study are (A) that authoritarian parenting pattern is negatively associated with school achievement; (B) that authoritative parenting style is positively associated with school achievement; and (C) that parental involvement affects children and their school performance. The results show a positive relation between authoritative parenting styles and children's school achievement, although the authoritarian style was associated with lower levels of school achievement. Additionally, the results indicate that the authoritarian style combined with involvement practices is a significant predictor of grades. The more authoritative means that parents use to socialize with their children, the more likely they are to achieve their parenting goals. Nevertheless, it is of critical importance for parents to focus on the learning process and not on the outcome.

**Keywords:** parenting styles; parenting practices; school achievement; elementary school

## 1. Introduction

This research examines the relationship between parenting patterns and children's school performance in Greek society, making it easy to understand that there is a high risk of negative parental involvement, behavior, and practices that could undoubtedly affect children and their school performance. Children, due to their sensitive age, are struggling to manage parental intervention in their school and personal lives; as a result, they often react and are influenced negatively. It is time to realize that parents' behavior regarding children's school performance and grades has associated risks. There is a lack of studies directly focused on the link between risk and parental practices, as these practices affect mainly the child's behavior and their educational advancement in general.

According to the literature, parenting style and parenting practices are important indicators of children's wider accomplishments. The following studies have shown that parental behaviors apply differently to each ethnicity or culture. According to Shumow et al. (1998), African–American parents are inclined to be less permissive and harsher than white parents, while Kokkinos and Vlavianou (2019) cited that Greek parents are overly involved in their children's rearing and are overprotective

and severe. According to Darling and Steinberg (1993), parenting practices (PP) refer to particular behaviors that parents use in order to develop their children's social skills. For example, parents enact daily socialization practices (e.g., assisting with children's homework, participating in teacher–parent meetings, etc.) in order to help their children succeed in school. Regarding school outcomes, these practices can be distinguished into three constructs: parental involvement; parental goals, aspirations and values; and parental monitoring (Spera 2005). Several studies have shown that parental involvement is a complex construct that has been defined in several ways (Gugiu et al. 2019). Based on Grolnick and Pomerantz (2022), parent involvement in children's schooling facilitates children's motivation, engagement, and learning, especially when such involvement is autonomy-supportive and affectively positive. However, parent involvement can have costs for children when it is controlling and affectively negative (Grolnick and Pomerantz 2022).

According to Epstein (1996), there are two types of parental involvement practices: those initiated by parents and those initiated by schools. Parental initiative involvement practices (PIIP) refer to parents' efforts to directly get involved with school activities and decisions, such as assisting with children's homework or attending school activities. Studies have predicted positive relations between PIIP and school outcomes (Epstein and Sheldon 2002) and, more precisely, that adolescents will spend much more time on their homework when parents assist them (Muller and Kerbow 1993). Dettemers et al. (2019), found high-quality parental homework involvement to be positively associated with students' well-being at school and at home, as well as with students' achievement in mathematics and language.

School-initiated parental involvement (SIPI) refers to schools endeavoring to provide parents with information about the progress of their children or in relation to school procedures, events, etc. Furthermore, studies in this field have found that firm and consistent discipline practices are positively related to involvement practices, while inadequate and negligent discipline actions were negatively correlated with coercive practices. The term 'firm' usually has a negative connotation, but the difference lies in how it is expressed. Firm and kind parents and firm and harsh parents both expect compliance. However, the essence of being kind and firm refers to responsive and demanding parental practices, which encourage compliance with expectations that strengthen parent–child relations (Larzelere et al. 2013). Moreover, parents' involvement behaviors may inadvertently lower children's autonomy and self-directed motivation, resulting in children developing negative perceptions about themselves. This may, in turn, negatively affect their academic achievement (Moè et al. 2020).

Parental monitoring is another method for parents to be involved in their children's education. Supervising and being aware of progress at school and relations with peers as well as homework involvement are some of the actions which fall under parental monitoring (PM) practices. Studies have identified the positive association between monitoring children's everyday life (e.g., after school activities) with higher academic performance (Spera 2005). Likewise, Kristjánsson and Sigfúsdóttir (2009) consider that reasonable monitoring predicts advanced academic achievement.

Parental styles (PSs) are essential variables that have been linked to school success. PSs are a concept first introduced by Baumrind (1971). Baumrind used the term 'parenting style' to describe beliefs and values regarding the child-rearing process, which disclose a parent's emotions for their children, the children's nature, and child-rearing practices. Darling and Steinberg (1993) defined PSs as expressions of parents' behaviors that establish an emotional climate during child-rearing; essentially, they are parental characteristics that remain stable over time and cultivate the emotional context for unfolding parenting practices. Complementing this approach, Kuppens and Ceulemans (2019) adopted a more person-centered approach, which engaged in patterns within individuals and suggested that PS

accounts for assorted parenting practices within the same person at the same time. Dettemers et al. (2019), declared that the parental provision of autonomy and competence support tends to satisfy the basic needs of their children (autonomy and competence) and, in turn, it might thus result in improved well-being. Baumrind (1971), based on extensive interviews and observations, suggested that there are three types of parenting styles: the authoritative style, which is characterized by warmth, caring, and responsiveness; the permissive style, which is expressed as indulgent and warm; and the authoritarian style, which is expressed with high levels of parental control and poor responsiveness. Complementary to Baumrind's theory, Maccoby and Martin (1983) introduced a four-style typology; essentially, they also added the neglectful style. As stated above, the main characteristics of the authoritative parenting style are responsiveness and warmth. This type of parenting has high maturity demands but provides a lot of support and affection when fostering children in pursuing and exploring their interests. These parents tend to communicate and explain their expectations and behaviors while encouraging children's independence. Authoritative parenting has been related to positive outcomes in children such as resilience, social competence, self-esteem, optimism, and academic achievement (Masud et al. 2015). A study examining the role of Greek fathers in children's psychosocial development found that children who considered their fathers to be authoritative showed greater levels of self-esteem and empathy when compared to children who described their fathers as authoritarian (Antonopoulou et al. 2012). Conversely, according to Steinberg (2001), an authoritarian style has consistently been associated with negative developmental outcomes in youths; this parental style may cause several behavioral and psychological problems in adolescents, such as anxiety, depersonalization (Wolfradt et al. 2003), and aggression (Hoeve et al. 2011). This style of parenting is neither responsive nor warm to the child; such parents tend to be strict and intolerant of selfishness. They expect obedience and they do not hesitate to assert power when they consider that their child has misbehaved (Baumrind 1978). Authoritarian parents have high expectations and maturity demands and express them through orders and rules without communicating the rationale behind these orders to their children (Maccoby and Martin 1983). In a study among the Mexican population, Calzada et al. (2015) revealed that authoritarian practices were highly associated with children internalizing and externalizing behavioral dysfunction at home. These significant findings stem from Hyojung et al. (2012) and Murray and Mulvaney (2012), who predicted an association between an authoritarian parenting style and higher scores in math and reading. The third parenting style, according to Baumrind (1978), is the permissive parent, defined as being moderate in responsiveness toward children; they tend to be unconcerned, dismissive, and lax in their tolerance of children's misbehavior and expectations. Studies on permissive (or indulgent) parenting have shown that children subjected to this style are more likely to suffer from depression, externalizing problems, school misconduct, lack of self-confidence, and poor social competence (Wolfradt et al. 2003). Additionally, parental experiences of guilt and shame regarding their children's academic efforts are likely to be an influential factor shaping parental style (Moè et al. 2020). In their study, Steinberg et al. (1992) examined relations between parenting style, parental involvement, and children's academic achievements. They found that children's academic functioning was positively related to authoritative parenting, although this relation increases when mediated by parental involvement. Furthermore, they highlighted the reverse relation of involvement and parenting style, finding that the less authoritative parents are, the less beneficial the influence of involvement in school performance is. One year later and based on the above evidence, Darling and Steinberg (1993) proposed the contextual parenting model. This model suggests that the effectiveness of children's educational goals set by parents (e.g., higher grades) depends on family climate (how parents encourage and support their children), which refers to the overall parenting style. However, a crucial

parameter in school achievement is undoubtedly the intensity of parental involvement. Grolnick (2016) notes that, although parents' direct involvement may predict children's success in school, these children were found to be more perfectionistic, less creative and with a higher depression risk. Indeed, highly controlling parenting undermines children's autonomy and reduces their intrinsic motivation to do well in school. The author noted that parents foster motivation when they value their children's efforts and performance. Equally, the atmosphere that parents create around their children's schoolwork may provide plenty of opportunities to either enhance or hamper children's motivational and academic development (Moè et al. 2020). Grolnick (2016) concluded that, instead of a strong control over children's performance, it is of critical importance for parents to focus on the learning process and not on the outcome. Moreover, a recent meta-analysis concluded that parents' support for homework is negatively associated with their children's school achievement. (Barger et al. 2019).

Finally, the neglectful parenting style—which is the most under-researched—is negatively linked to children's development. As per Kuppens and Ceulemans (2019), neglectful parenting is expressed by low responsiveness and low demandingness. Hoeve et al. (2011) have shown that neglectful parents rear children with a lack of self-regulation and self-reliance, low social and school competence, and high levels of stress and antisocial behavior.

### 1.1. School Achievement

School achievement is considered to be a significant predictor of a student's future social status and professional career (Sijtsema et al. 2014). According to Helmke and Schrader (2001), school achievement (SA) can be defined as "cognitive learning outcomes" that are "products of instruction or aimed at by instruction within a school context". Many studies have shown that parental practices can either facilitate or hinder children's achievement, depending on whether the parents' foster autonomy or control in their children, respectively (Moè et al. 2020). Additionally, previous studies have shown that a range of variables determines school achievement, such as parenting style, parenting practices (Love et al. 2020; Korucu et al. 2020), family characteristics, aspects of the school context (Karibayeva and Bogar 2014), and students' personal characteristics, such as individual differences in temperament (Checa and Abundis-Guitiérrez 2017), socioeconomic backgrounds, and ethnic groups (Zahedani et al. 2016). Each of these variants has an important impact on the development of SA.

### 1.2. The Role of Gender

Various studies have documented that girls outperform boys in terms of school achievement (Duckworth and Seligman 2006; Voyer and Voyer 2014). Although the causes of gender differences in school attainment have not been scientifically clarified, several authors had provided some theoretical validations. Hicks et al. (2008) stated that genetic and biological determinants attribute to gender differences, while Spinath et al. (2014) found variations in competencies and abilities to be responsible. Freudenthaler et al. (2008) cited differences in personality characteristics for disparities in social experiences, and variations in learning styles have also been considered as possible explanations.

Downey and Vogt Yuan (2005) found that school achievement depended on the type of school subject, with boys performing better in mathematics and girls performing better in languages. However, Burusic et al. (2012) stated that recently girls are ranked as more successful than boys in 'masculine subjects' such as science and mathematics. In a study conducted in Croatia with a huge sample of fourth- and eighth-grade students from 844 elementary schools, researchers aimed to examine the teacher–children gender-interaction outcomes on children in relation to school achievement. The authors found that, in general, girls outperformed boys through their grades. Additionally, fourth-grade girls in particular achieved better results in

language, nature, society, and math subjects. Regarding the eighth-grade students, girls achieved better results on language, biology, and chemistry tests, while boys scored better on geography and physics tests. No significant differences were recorded in relation to history tests, and no association was observed between the teachers' gender and school achievement (Burusic et al. 2012). A current major debate is also the influence of teachers on gender-typing. It has been confirmed that the school system tends to frown upon the independent, assertive, competitive, and boisterous qualities that parents and the culture have encouraged in boys from infancy. Girls, who are more verbally orientated, generally better-behaved, and better at following rules, typically experience greater acceptance from teachers who—at least in the early grades—are likely to be female. Therefore, it is not surprising that, from the start, girls tend to like school more than boys and also perform better in their academic work. Boys create more problems for teachers and elicit more criticism from them, and they often perform at a lower level than their female classmates. These gender differences in classroom interactions and attitudes may go some way to explaining variations in academic performance between boys and girls across the school system (Ruble et al. 2006).

### 1.3. The Role of Cultural Differences

Quite a few studies have mentioned the effect of ethnicity on parenting practices. Shumow et al. (1998) found that African–American parents tend to be less permissive and harsher than white parents. It has been found that authoritative parenting, along with positive practices, is of significant importance for Latino–American and Asian–American students in comparison to African–American and European–American students. Conversely, European–American students are more influenced by emotional support than Latino–American and Asian–American students (Rosenzweig 2001); this may also be the reason for the fact that Chinese–American parents were reported to be stricter than European–American parents (Lin and Fu 1990). For instance, respect and duty are fundamental elements of a Mexican family (Lindsey 2018) and, according to Calzada et al. (2015), hierarchy and family loyalty are placed above an individual's desires. In other words, Latino parents require deference to adults and obedience, which cultivates and develops an authoritarian parenting style (Calzada et al. 2015). A recent study conducted by Filus et al. (2019) in four different European countries (Norway, Switzerland, Greece, and Poland), each with different cultures, living conditions, and values, found that—conversely to Norwegian, Swiss and Polish fathers—the autonomy-granting and responsiveness of Greek fathers was negatively associated with functional and psychological connectedness. Filus et al. (2019) claimed that autonomy-granting plays a significant role in adolescents' individuation. Individuation is a crucial function for life outcomes and is highly associated with emotional adjustment and academic achievement. In a comparative study conducted in Italy, Greece, and Sweden, Olivari et al. (2015) sought to explore differences and similarities among adolescent perceptions of parenting styles. They found that, in all three countries, the dominant style was authoritative. In comparison, Italian parents scored higher in authoritarianism, followed by Greek parents. Meanwhile, Swedish and Greek adolescents perceived their parents as being more permissive than Italian parents. According to Zervides and Knowles (2007), Greek culture promotes family loyalty, the cultivation of harmonical relationships among group members, and devotion to group norms. Obedience and conformity to parental rules have been linked with child-rearing austerity. In addition, Kokkinos and Vlavianou (2019) found that Greek parents are overprotective and their involvement in child-rearing is excessive; they also highlight that Greek culture promotes severe and controlling parenting practices.

### 2. Materials and Method

### 2.1. Participants and Instruments

The participants in this study included 101 parents (61 mothers and 40 fathers) residing in Greece. The prerequisite for inclusion was that participants had to have at least one child attending the fourth, fifth, or sixth grade of elementary school. The participants voluntarily answered an online questionnaire, which was created using the Google Forms platform, and informed consent was mandatory. This research only included the above grades, as in the first, second, and third grades of elementary school only an oral assessment is provided by the class teacher, who informs the students' parents about their progress.

### 2.2. Procedures and Data Analysis

In line with previous findings, the aim of this study was to examine associations between parenting and primary school students' achievements in Greece. The initial assumption was that both parenting practices and parenting style are associated with academic achievement. Thus, the primary hypotheses under examination in this study were (A) that an authoritarian parenting pattern is negatively associated with school achievement; (B) that an authoritative parenting style is positively associated with school achievement; and (C) that parental involvement affects children and their school performance. With the intention of assessing parental involvement, supervision, and monitoring, this study used the 24-item parenting practices inventory as revised by Rodríguez and Rosquete (2019). The questionnaire consisted of four subscales (firm and consistent discipline; involvement practices; negligent and inadequate practices; and coercive practices) and responses were classified using a five-point scale (1 = never or almost never to 4 = very frequently). Additionally, the parenting-style questionnaire (PSQ) developed by Robinson et al. (1995) was used to assess three parenting typologies: authoritative, permissive, and authoritarian. The PSQ consisted of 62 items; 27 items were in relation to authoritative style (alpha 0.91), 20 items were in relation to authoritarian style (alpha 0.86), and 15 items were in relation to permissive style (alpha 0.75). Parents were asked to rate each statement on a five-point Likert scale (1 = never to 5 = always). For the purposes of this study, the 30-item short version of the PSQ was used. The authoritarian style consisted of thirteen items, the authoritative style consisted of thirteen items and the permissive style consisted of four items. While school achievement (SA) was measured for the previous academic year (September to June), grades were used as the basis for measuring academic achievement. Our data consisted of the average score of marks in languages, mathematics, and history. Parents were required to fill out the grades based on a zero to ten (0–10) scale in relation to the aforementioned subjects for each semester. At first, the correlation between variables was tested to verify if they were suitable for a mediation analysis. A series of independent t-tests examined possible gender differences between parents in parental style, parental practices, and school achievement. Pearson correlation coefficients were used to calculate relations between the variables. For the result analysis, the SPSS for Windows statistics package was used. In relation to the structural equation model, all latent variables were significant, and the test presented good fit indexes.

### 2.3. Results

A total of 101 volunteer parents (61 mothers and 40 fathers) answered the self-reported questionnaires. The participants' mean age was M = 41.83, SD = 6.82; the fathers' mean age was M = 42, SD = 6.1; while the mothers' mean age was M = 41.1, SD = 6.82. The children's mean age was M = 11.03, SD = 0.82. Parents' education levels consisted of 15.8% (*n* = 16) that held a master's degree, 6.9% (*n* = 7) that held a doctorate, 29.7% (*n* = 30) that held a bachelor's degree, 31.7% (*n* = 32) that held a high school degree, and 7.9% (*n* = 8) that had less than a high school diploma. The parenting

practices inventory (PPI) was divided into four subscales. Each subscale was labelled in relation to the type of practice: firm and consistent discipline practices (FCDP, $\alpha$ = 0.777); involvement practices (IP, $\alpha$ = 0.908); negligent and inadequate practices (NIP, $\alpha$ = 0.665); and coercive practices (CP, $\alpha$ = 0.791). Additionally, parenting style (PS) consisted of three subscales: authoritative style ($\alpha$ = 0.971); authoritarian style ($\alpha$ = 0.924); and permissive style. The initial analysis of permissive style was predicted to be $\alpha$ = 0.490; therefore, one item, "I spoil my child," was removed. Therefore, the permissive style included three items resulting in $\alpha$ = 0.512. Both patterns resulted in configuration invariance and were statistically significant ($p \leq 0.001$). There were linear relationships between variables with no multi-collinearity and there was homogeneity of variance covariance matrices (Table 1).

**Table 1.** Mean scores of parenting typologies.

|  | Mean | Std. Deviation | N |
|---|---|---|---|
| Grade | 9.1241 | 0.71505 | 101 |
| Authoritative | 3.8380 | 0.94968 | 101 |
| Permissive | 2.6879 | 0.72637 | 101 |
| Authoritarian | 2.3928 | 0.76786 | 101 |
| Firm and Consistent | 2.8528 | 0.63587 | 101 |
| Involvement | 2.6277 | 0.71513 | 101 |
| Negligent and Inadequate | 2.0745 | 0.51129 | 101 |
| Coercive | 1.5904 | 0.61985 | 101 |

A correlation analysis showed that there was a significant positive relationship between authoritative parenting style (r = 0.417, $p \leq 0.001$) and school achievement (Table 2). Firm and consistent practices (r = 0.411, $p \leq 0.001$) and involvement practices (r = 0.431, $p \leq 0.001$) were also positively related with school achievement (Table 3). On the other hand, an authoritarian style recorded significant negative association with school achievement (r = −0.426, $p \leq 0.001$). Similarly, a permissive style (r = −0.258, $p$ = 006) and negligent/inadequate (r = −0.258, $p$ = 006) and coercive practices (r = −0.288, $p$ = 0.002) showed a negative relationship with grades at school (Table 4).

**Table 2.** Correlation between involvement and authoritarian style and school achievement.

|  | Model | Unstandardized Coefficients | | Standardized Coefficients | T | Sig. | 95.0% Confidence Interval for B |
|---|---|---|---|---|---|---|---|
|  |  | B | Std. Error | Beta |  |  | Lower Bound |
| 1 | (Constant) | 7.991 | 0.256 |  | 31.211 | 0.000 | 7.482 |
|  | Involvement | 0.431 | 0.094 | 0.431 | 4.586 | 0.000 | 0.244 |
| 2 | (Constant) | 8.982 | 0.442 |  | 20.309 | 0.000 | 8.104 |
|  | Involvement | 0.293 | 0.104 | 0.293 | 2.812 | 0.006 | 0.086 |
|  | Authoritarian | −0.263 | 0.097 | −0.282 | −2.706 | 0.008 | −0.456 |

**Table 3.** Correlation between involvement and authoritarian style and school achievement.

| | Model | Unstandardized Coefficients | | Standardized Coefficients | T | Sig. | 95.0% Confidence Interval for B |
|---|---|---|---|---|---|---|---|
| | | B | Std. Error | Beta | | | Lower Bound |
| 1 | (Constant) | 7.991 | 0.256 | | 31.211 | 0.000 | 7.482 |
| | Involvement | 0.431 | 0.094 | 0.431 | 4.586 | 0.000 | 0.244 |
| 2 | (Constant) | 8.982 | 0.442 | | 20.309 | 0.000 | 8.104 |
| | Involvement | 0.293 | 0.104 | 0.293 | 2.812 | 0.006 | 0.086 |
| | Authoritarian | −0.263 | 0.097 | −0.282 | −2.706 | 0.008 | −0.456 |

| | Model | 95.0% Confidence Interval for B |
|---|---|---|
| | | Upper Bound |
| 1 | (Constant) | 8.499 |
| | Involvement | 0.618 |
| 2 | (Constant) | 9.861 |
| | Involvement | 0.500 |
| | Authoritarian | −0.070 |

**Table 4.** Predictive model of involvement practices and authoritarian style.

| Model | R | R Square | Adjusted R Square | Std. Error of the Estimate | Change Statistics | | | |
|---|---|---|---|---|---|---|---|---|
| | | | | | R Square Change | F Change | df1 | df2 |
| 1 | 0.431 [a] | 0.186 | 0.177 | 0.64861 | 0.186 | 21.029 | 1 | 92 |
| 2 | 0.497 [b] | 0.247 | 0.230 | 0.62741 | 0.061 | 7.321 | 1 | 91 |

**Model Summary** [c]

| Model | Change Statistics | Durbin–Watson |
|---|---|---|
| | Sig. F Change | |
| 1 | 0.000 | |
| 2 | 0.008 | 1.513 |

[a] Predictors: (Constant) Involvement, [b] redictors: (Constant), Involvement andAuthoritarian, and [c] Dependent Variable: School Grades.

According to Table 3, the coefficients indicate that an authoritarian style combined with involvement practices is a fair predictor of school achievement ($\beta = 8.9$, $p \leq 0.001$). An authoritarian style has a negative effect on school achievement ($\beta = -0.263$, $p = 0.008$), while involvement practices show a positive relationship with school grades ($\beta = 0.6$, $p \leq 0.001$). An application of an authoritarian style and involvement practices appears to have the strongest association ($R = 0.50$) for predicting grade outcomes. The proportion of variation in the outcome variable was $R^2 = 25$, thus 24.7% of the variance in the data can be explained by the predictor variable. An authoritarian style with an involvement practices model was a significant predictor of exam performance, $F(2,91) = 14.897$, $p < 000$.

The results revealed that girls (M = 9.3, SD = 0.62) performed slightly better than boys (M = 8.8, SD = 0.65) based on their average mean grades (Table 5). An additional independent t-test analysis was applied in order to test the significance of differences between gender and subject grades (t (94) = −3.54, *p* <0.001). In testing the correlation between parenting style and gender (Tables 6 and 7), the findings revealed that mothers scored higher in authoritativeness than fathers (mothers M = 4.04, fathers M = 3.38).

**Table 5.** Means and standard deviations of subjects by gender.

|  | Child's Gender | N | Mean | Std. Deviation | Std. Error Mean |
|---|---|---|---|---|---|
| Language Grade | Male | 47 | 8.96 | 0.779 | 0.114 |
|  | Female | 50 | 9.36 | 0.663 | 0.094 |
| Math Grade | Male | 47 | 8.87 | 0.875 | 0.128 |
|  | Female | 50 | 9.30 | 0.789 | 0.112 |
| History Grade | Male | 47 | 8.87 | 0.778 | 0.115 |
|  | Female | 50 | 9.38 | 0.725 | 0.103 |
| Group Statistics |  |  |  |  |  |
|  | Child's gender | N | Mean | Std. Deviation | Std. Error Mean |
| School Grades | Male | 47 | 8.8840 | 0.65623 | 0.09676 |
| Total | Female | 50 | 9.3467 | 0.62437 | 0.08830 |

**Table 6.** Means and standard deviations of parenting style by gender.

| Parent Gender |  | Permissive | Authoritative | Authoritarian |
|---|---|---|---|---|
| Male | Mean | 2.6396 | 3.3867 | 2.4286 |
|  | N | 40 | 40 | 40 |
|  | Std. Deviation | 0.62053 | 1.01881 | 0.79269 |
| Female | Mean | 2.7596 | 4.0410 | 2.4257 |
|  | N | 61 | 61 | 61 |
|  | Std. Deviation | 0.76771 | 0.79342 | 0.72979 |
| Total | Mean | 2.6997 | 3.8269 | 2.3942 |
|  | N | 101 | 101 | 101 |
|  | Std. Deviation | 0.71259 | 0.94486 | 0.76013 |

**Table 7.** Means and standard deviations of parenting practices by gender.

| Parent Gender |  | Firm and Consistent Discipline | Involvement | Negligent and Inadequate | Coercive |
|---|---|---|---|---|---|
| Male | Mean | 2.6892 | 2.4696 | 1.9459 | 1.6892 |
|  | N | 40 | 40 | 40 | 40 |
|  | Std. Deviation | 0.67833 | 0.80278 | 0.46821 | 0.67804 |
| Female | Mean | 2.8880 | 2.6598 | 2.2404 | 1.5533 |
|  | N | 61 | 61 | 61 | 61 |
|  | Std. Deviation | 0.56415 | 0.61793 | 0.52133 | 0.55736 |
| Total | Mean | 2.8416 | 2.6163 | 2.1089 | 1.5866 |
|  | N | 101 | 101 | 101 | 101 |
|  | Std. Deviation | 0.62910 | 0.70915 | 0.53252 | 0.60564 |

### 3. Discussion

This study distinguished two parenting typologies and examined the manner in which both parenting style and parenting practices are associated with children's school achievement. An additional hypothesis assumed that authoritative parenting is positively related with school grades, while authoritarian parenting negatively affects grades. Moreover, parenting involvement was hypothesized to affect children's school performance in a positive way. Consistent with previous studies, authoritative parenting was found to be the strongest predictor for higher school achievement. Moreover, for parenting practices, parental involvement represented the highest positive correlation. Studies have shown that when parents follow supporting practices, such as assisting with children's homework or activities, this results in positive outcomes on children's academic achievement. Opposing, beating, or criticizing children may cause behavioral problems (aggressiveness, disobedience, etc.), which decrease academic performance (Murray-Harvey and Slee 2007). Moreover, Checa and Abundis-Guitiérrez 2017) found a negative relationship between academic performance and a coercive parenting style.

However, some studies showed associated higher scores in reading and math with an authoritarian style (Hyojung et al. 2012; Murray and Mulvaney 2012). A possible explanation, which Murray and Mulvaney (2012) provided for their study results, is that there are probably some participants that fit into the 'authoritarian' style who were not lacking warmth or being as controlling as authoritarians. Meanwhile, Garcia and Gracia (2009) showed that indulgent/permissive parenting was related to better grade outcomes among a large sample of Spanish adolescents. Garcia and Gracia (2009) also found that the influence of each parental style is a matter of culture and ethnicity and cannot be generalized for all societies. There is a clear boundary between Eastern and Western cultures. For instance, authoritarian parenting does not harm children's mental health in Arab or Asian societies as it does in Western societies. Matejevic et al. (2014) also found that culture is a dominant aspect between parenting styles and academic achievement.

According to Barger et al. (2019), parents' involvement was positively related to children's social and emotional adjustment and negatively related to their delinquency. Analyses focusing on children's academic adjustment revealed that different types of involvement were similarly positively associated with such adjustment (Barger et al. 2019). Dettemers et al. (2019) have called attention to parental involvement's relevant role in children's schooling and success. Concerning achievement, results in the same research were in line with previous studies, providing evidence of a positive relationship between parental involvement in schooling and students' achievement.

One noticeable outcome was also the correlation between firm and consistent practices and an authoritarian style. Cavell (2002) supported the view that, when rules and boundaries are clearly specified, parents act better because a family system is more functional when each member is treated thoughtfully and obeys household rules (Gladding 1998). The results indicated that mothers were more authoritative than fathers. Matejevic et al. (2014) supported the view that an authoritative style is a mother's characteristic, while fathers tend to be more authoritarian due to their inner need for autonomy. Moreover, contradictory parenting styles among parents operate in complementarity ways to their parental roles. McKinney and Renk (2008) showed that fathers and mothers can employ different parenting styles, and children that grow up in two-parent households are influenced by both parents' practices (Martin et al. 2007). In a joint-parenting family, the less-effective style can be buffered by the effective one, as was suggested by McKinney and Renk (2008).

An observable finding was the high-grade mean (M = 9.1) across the whole sample. The explanation for these results is that grades in elementary school are based on a ten-point scale. In greater school levels where the grade scale goes up to twenty, the range of school marks would be broader, showing greater variance for each subject. Complimenting the above-mentioned and similar studies, recent studies show that girls performed slightly better in all three subjects than boys (Duckworth and Seligman 2006;

Freudenthaler et al. 2008). Although the grade differences were not significant, according to Montroy et al. (2014), outcome differences are not recognizable in the first years of primary school as they usually appear in middle- and high-school children. The phenomenon that girls outperform boys in most school subjects has been examined in several countries (e.g., USA: Epstein (1996), Belgium, Hong Kong and Cyprus). Studies have shown that girls tend to be more disciplined than boys. Duckworth and Seligman (2006) cited that girls devote double the time to their homework. In their study, Freudenthaler et al. (2008) observed that conscientiousness, which is a personality trait compared with intelligence (Poropat 2009), was only positively connected with school performance in girls. The conscientiousness of girls, who performed better according to Maltby et al. (2010), is a powerful predictor for school achievement. Younger et al. (1999) conducted a study on gender differences and classroom interactions; their findings revealed that teachers evaluated girls as more articulate and organized. Additionally, girls enforced advanced learning strategies and participated more in classroom interactions.

Since the characteristics comprising parenting practices and parenting style were consistent with relative research in this field, this study's conclusions met the purpose of assessing parenting in Greece. Based on Gordon and Cui (2015), school and academic achievement is related to success, satisfaction, and career autonomy in adulthood. Likewise, supporting primary-school children in achieving academic success is vital, not only to stimulate their academic performance and abilities but also to reinforce their social skills. Several authors pointed out that authoritative parenting affects psychosocial maturity, and psychosocial maturity determines how children perform in school. Equally, psychosocial maturity is dignified by work orientation, self-reliance, and self-identity, which in turn are associated with higher school grades (Kordi and Baharudin 2010). According to Barger et al., parents' homework assistance was negatively associated with children's achievement, but not engagement or motivation (Barger et al. 2019).

## 4. Conclusions

This study has drawn the following conclusions: both parenting style and parenting practices influence children's school achievements, which underpins the powerful impact of parenting on children's development. The study findings appear to confirm the assertion that, akin to Baumrind's theory, authoritative parenting confers school advantages on Greek children. The results point out that, in Greek society, authoritative parenting seems to be the optimal parenting style and authoritarian is the poorest parenting style. Firm and consistent discipline combined with parental involvement was the predominant aspect for parenting practices. Similar to previous findings, this study strengthens the demand for public consideration as to the value of parenting in child-rearing. The findings indicate that, for Greeks, an authoritarian style is the most actual and effective one, while parental involvement is a significant matter. Moreover, gender differences were assessed and revealed that mothers score slightly higher in authoritativeness than fathers and girls outperform boys in all school subjects. Based on the above, we suggest social policy should develop more intervention strategies and parenting coaching programs, in order to train and counsel parents in their important mission. Finally, although this study's conclusions met the purpose of determining parenting in Greece, more research is required to test the causal effects of parenting in Greek society.

*Limitations*

The current research has several limitations. Although the sample was realistically diverse, it was limited to students attending fourth, fifth, and six elementary grades. A larger-scale study entailing more participants would be preferable and representable. In future research, it would be worthwhile to use a longitudinal design involving a wider range of schoolchildren. A limitation of this study was that school grades were self-reported. There was a hidden assumption that parents had complete information and that they provided it. Another limitation can refer to possible social desirability bias in the

consent form of the questionnaire, specifically in the section stating that, "Parenting plays an indispensable role in child development." An additional limitation is the cross-sectional nature of the study and its participant number, which prevents the generalizability of results and the assumption of casual relationships between the investigated variables. Although this research controlled for a number of parenting patterns as potential confounders, there are likely residual, confounding dynamics for which evidence was not available as to the interplay of children's motivation with parenting. Several studies had predicted that motivation is linked with school achievement and is mediated by parenting (Józsa et al. 2019; Gonzalez and Wolters 2006). As children's socialization depends on their parents, according to Chen et al. (2019), and children's and parents' perceptions and beliefs differ, these differences lead to diverse implications for behavior and motivation. Further research is needed to examine additional, complex models of personality and their interactions with school performance. In the same manner, children's temperaments and personality characteristics in general were not taken into consideration. Dimensions of personality variables could have been essential because, as Barchard and Christensen (2007) supported, personality and intelligence could predict better academic achievement. Lastly, while the method is reliable enough, it has several limitations and a broader and more current revision of methodology would be given more weight to the validity of the procedure.

Some future research that could be conducted could have a larger sample size of participants in order to control the risk of reporting false-negative or false-positive findings, thus leading to more accurate and representative results; Finally, the impact of different ethnicities also needs to be assessed in order to achieve a respective comparison.

**Author Contributions:** Conceptualization, D.T. and R.D.T.; methodology, I.G.L.; software, I.G.L.; validation, D.T., R.D.T. and I.G.L.; formal analysis, R.D.T.; investigation, R.D.T.; resources, D.T.; data curation, I.G.L.; writing—original draft preparation, R.D.T.; writing—review and editing, D.T.; visualization, D.T.; supervision, I.G.L.; project administration, R.D.T. All authors have read and agreed to the published version of the manuscript.

**Funding:** This research received no external funding

**Institutional Review Board Statement:** The study was conducted in accordance with the Declaration of Helsinki, and approved by the Institutional Review Board.

**Informed Consent Statement:** Informed consent was obtained from all subjects involved in the study.

**Data Availability Statement:** The data that support the findings of this study are available on request from the corresponding authors

**Conflicts of Interest:** The authors declare no potential conflicts of interest with respect to the research, authorship, and/or publication of this article.

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
