# Peer review of "Relations between Parenting Style and Parenting Practices and Children’s School Achievement"

_socsci, doi:10.3390/socsci12010005_

Round 1

Reviewer 1 Report

The article is well structured but does not make a relevant contribution to education. It is based on more than well-known theories, with references that are not very up to date. It lacks a more rigorous methodology, as the instrument used is a self-perception questionnaire via Google, which lacks control of other variables, such as social desirability.  The results and conclusions are clear, but do not add value. 

Author Response

Thank you for your patient review. The below references have also been added.

Dettemers, S.; Yotyodying, S.; Jonkmann, K. Antecedents and Outcomes of Parental Homework Involvement: How Do Family-School Partnerships Affect Parental Homework Involvement and Student Outcomes? Front. Psychol. 2019, 10, 1–13

Gugiu, P.C.; Gugiu, M.R.; Barnes, M.; Gimbert, B.; Sanders, M. The Development and Validation of the Parental Involvement Survey in their Children’s Elementary Studies (PISCES). J. Child Fam. Stud. 2019, 28, 627–641

Moè, A., Katz, I., Cohen, R., & Alesi, M. (2020). Reducing homework stress by increasing adoption of need-supportive practices: Effects of an intervention with parents. Learning and Individual Differences, 82, 101921.

Reviewer 2 Report

The manuscript deals with an interesting topic. Introduction, methods and discussions may be further improved, but the findings are interesting. I hope my comments can help the author(s) in further improve their manuscript.

Throughout the manuscript, grammar checks and language revisions are required. I will not list all, but a few examples are Abstract. A) and B) parenting should be followed by style/pattern; C) parental involvement instead of parenting involvement; grades instead of ‘grade performance’; Materials and Methods, line 4, answered to instead of on an online questionnaire; the test presented good fit indexes instead of statistics.

Introduction

I wonder if the references to permissive and neglectful parenting styles are necessary, seeing that the author(s) did not take into consideration these as variables in their investigation.

page 3, line 14, I do not understand the use of the term ‘current’ in that sentence. To what findings do the author(s) refer? The finding of the study in their paper or to Wolfradt et al 2003; Williams et al 2009 – could not be referred to as ‘current’.

The authors might want to improve the introduction and discussions by discussing some recent pieces of evidence on the topic of their paper. For example

A recent meta-analysis concluded that parents’ support for homework is negatively associated with their children’s school achievement

Barger, M. M., Kim, E. M., Kuncel, N. R., & Pomerantz, E. M. (2019). The relation between parents’ involvement in children’s schooling and children’s adjustment: A meta-analysis. Psychological bulletin, 145(9), 855.

Still on parental involvement:

Grolnick, W. S., & Pomerantz, E. M. (2022). Should parents be involved in their children’s schooling? Theory Into Practice, 61(3), 325-335.

Moè, A., Katz, I., Cohen, R., & Alesi, M. (2020). Reducing homework stress by increasing adoption of need-supportive practices: Effects of an intervention with parents. Learning and Individual Differences, 82, 101921.

Maybe doing a quick hint to parental involvement during the Covid era would also be the case. For example:

Ribeiro, L. M., Cunha, R. S., Silva, M. C. A. E., Carvalho, M., & Vital, M. L. (2021). Parental involvement during pandemic times: Challenges and opportunities. Education Sciences, 11(6), 302.

In general, there’s a paucity of references, mainly outdated (and the citation format is heterogeneous. Please, refer to authors' guidelines and make them homogeneous). Also, some of the papers cited within the manuscript are actually not mentioned in the references list.

The Role of Gender

Among other explanations provided, I would also debate the influence of teachers on gender-typing

Although teachers often seem to pay more attention to boys than to girls, the general culture of the classroom and the school in some ways favour girls. The school system tends to frown upon the independent, assertive, competitive and boisterous qualities that parents and the culture have encouraged in boys from infancy. Girls, who are more verbally orientated, generally better behaved and better at following rules, typically experience greater acceptance from teachers who – at least in the early grades – are likely to be female. It is not surprising, then, that from the start girls tend to like school more than boys and to perform better in their academic work. Boys create more problems for teachers and elicit more criticism from them, and they often perform at a level that is not only lower than their female classmates. These gender differences in classroom interaction and attitudes may go some way to explaining variations in academic performance between boys and girls across the school system (e.g. Leman, 2004; Ruble, Martin, Berembaum, 2006; Sulla, Pasetti, Dall’Olio, 2022).

Leman, P. J. (2004) And your chosen specialist subject is... The Psychologist, 17(4), 196–198.

Ruble, D., Martin, C. & Berenbaum, S. (2006) Gender development. In W. Damon & R.M. Lerner (Series eds) & N. Eisenberg (Vol. Ed.), Handbook of child psychology: Vol. 3. Social, emotional, and personality development (6th ed, pp. 858–932) New York: Wiley.

Sulla, F., Pasetti, A., & Dall'Olio, I. (2022). Processi di tipizzazione di genere in famiglie con genitori migranti e a rischio di povertà educativa: un’esperienza formativa condotta nell’ambito del Progetto “Ali per il Futuro” [Gender-typing processes in families with migrant parents at risk of educational poverty: a training experience conducted as part of the "Ali per il Futuro" project]. Rivista italiana di educazione familiare, 20(1), 33-46.

The author(s) might want to differentiate the last paragraph from the previous one (The Role of Gender). I wonder if is the role of cultural differences or the effect of ethnicity being questioned there?

Materials and Methods

How come the author(s) decided to include only 4th to 6th graders? I suggest justifying your choice.

The author(s) should also declare the study objectives/hypothesis here and not only in the abstract.

A division of this paragraph into Participants, Instruments, Procedures, and Data Analysis would make it easier to follow. Results should be presented in a different section.

The author(s) declared the use of the Parenting Practices Inventory (PPI) in the version of Soriano Rodrìguez et al., 2019. An adaptation and validation to the Greek context exist? Otherwise, did you use a mere translation of the items? If this is the case, I suggest declaring it within the limitations of the study. Same for the PSQ.

A t-test was used to examine gender differences in the variables of the study. Do the author(s) mean gender differences between children, parents or both?

Results presentation must be significantly revised as it is really confusing (this, and missing references made me think I might have read an early draft of the work. Is this the case?); results, in general, would need a revision by a statistician. For example text styles must be consistent throughout tables; reference to each table must be reported in the text; table captions must be consistent with table contents. The author(s) declared: ‘At first, the correlation between variables was tested to verify if they were suitable for a mediation analysis.’ It does not seem a mediation analysis was performed eventually. I suggest declaring this and also what kind of model was tested.

Limitations

I suggest putting this section after Conclusions and implementing suggestions for future investigations that would overcome the limitations reported.

Another limit is the cross-sectional nature of the study (and participant number), which prevents the generalisability of results and the assumption of casual relationships between the investigated variables. For the same reason, I suggest being cautious in declaring that ‘this study’s conclusions met the purpose of determining parenting in Greece’.

Author Response

Thank you very much for your review. According to your suggestions and comments, we have made the proper and necessary adjustments. Please see the attachment.

Reviewer 3 Report

Thematic is very interesting.

Please, add more recent references about this theme.

Author Response

Thank you for your comment and concern. The manuscript has been revised and the below references have been added accordingly.

Dettemers, S.; Yotyodying, S.; Jonkmann, K. Antecedents and Outcomes of Parental Homework Involvement: How Do Family-School Partnerships Affect Parental Homework Involvement and Student Outcomes? Front. Psychol. 2019, 10, 1–13

Gugiu, P.C.; Gugiu, M.R.; Barnes, M.; Gimbert, B.; Sanders, M. The Development and Validation of the Parental Involvement Survey in their Children’s Elementary Studies (PISCES). J. Child Fam. Stud. 2019, 28, 627–641

Moè, A., Katz, I., Cohen, R., & Alesi, M. (2020). Reducing homework stress by increasing adoption of need-supportive practices: Effects of an intervention with parents. Learning and Individual Differences, 82, 101921.

Round 2

Reviewer 1 Report

the article has improved, but has important limitations in methodology. The procedure of applying the questionnaire makes it very unreliable. The references have been updated, but a broader and more current revision would be needed to give more weight to the validity of the theory today, although it is widely accepted. The conclusions do not bring anything new to the field, which makes the article very uninteresting.

Author Response

Thank you for the time and effort that you dedicated to providing feedback on our manuscript and are grateful for your insightful comments on our paper. We have been able to incorporate changes to reflect most of the suggestions provided. While we appreciate your feedback, we think this study makes a valuable contribution to the field and has a pedagogical intrest because this study’s conclusions met the purpose of determining parenting in Greece and underline the importance for parents to focus on the learning process and not on the outcome. However more research is required to test the causal effects of parenting in Greek society.

Round 3

Reviewer 1 Report

Better explained this time